# Dynamic Metal Nanoclusters: A Review on Accurate Crystal Structures

**DOI:** 10.3390/molecules28145306

**Published:** 2023-07-10

**Authors:** Xiang Liu, Fan Peng, Gao Li, Kai Diao

**Affiliations:** 1Hunan Drug Inspection Center, Hunan Institute for Drug Control, Changsha 410013, China; 145408@hotmail.com; 2Public Course Teaching Department, Changsha Health Vocational College, Changsha 410013, China; pennymimi@hotmail.com; 3Dalian Institute of Chemical Physics, Chinese Academy of Sciences, Dalian 116023, China; 4College of Mathematics and Physics, Chengdu University of Technology, Chengdu 610059, China

**Keywords:** metal nanoclusters, dynamic structures, catalysis, synthesis

## Abstract

Dynamic metal nanoclusters have garnered widespread attention due to their unique properties and potential applications in various fields. Researchers have been dedicated to developing new synthesis methods and strategies to control the morphologies, compositions, and structures of metal nanoclusters. Through optimized synthesis methods, it is possible to prepare clusters with precise sizes and shapes, providing a solid foundation for subsequent research. Accurate determination of their crystal structures is crucial for understanding their behavior and designing custom functional materials. Dynamic metal nanoclusters also demonstrate potential applications in catalysis and optoelectronics. By manipulating the sizes, compositions, and surface structures of the clusters, efficient catalysts and optoelectronic materials can be designed and synthesized for various chemical reactions and energy conversion processes. This review summarizes the research progress in the synthesis methods, crystal structure characterization, and potential applications of dynamic metal nanoclusters. Various nanoclusters composed of different metal elements are introduced, and their potential applications in catalysis, optics, electronics, and energy storage are discussed. Additionally, the important role of dynamic metal nanoclusters in materials science and nanotechnology is explored, along with an overview of the future directions and challenges in this field.

## 1. Introduction

### 1.1. Research Background

In recent years, many fundamental properties of dynamic metal nanoclusters have been gradually revealed. Due to their unique optical and electronic properties, these materials have found wide-ranging applications in assembly, biological labeling and sensing, drug delivery and therapy, molecular recognition and molecular electronics, photoluminescence, and catalysis. Their applications have been extensively demonstrated [1,2,3,4,5,6,7]. Under ligand protection, nanostructured metal materials composed of Au and Ag with precise structures have become typical prototypes. The correlation between structural properties at the atomic level and their assembly mechanisms has been extensively investigated and understood [8,9,10,11,12].

In addition to well-established size-focused methods, single-atom substitution and size/structure conversion have also been considered successful approaches for preparing atomically precise clusters [13,14,15,16,17,18,19,20]. The single exchange of metal atoms in and out of metal clusters is of significant importance for establishing structure–property relationships based on the single-variant principle. Wang et al. [21] proposed a highly controlled method to shuttle single Ag or Cu atoms into the central void of Au_24_ nanoclusters, resulting in Ag_1_Au_24_ and Cu_1_Au_24_ nanoclusters. The ligand exchange-induced size/structure transformation (LEIST) [16,17,18,19], structure transformation induced by heteroatom doping, and oxygen-induced size/structure transformation [20,22,23,24] are considered successful methods for the preparation of atomically precise nanoclusters with precise structures. The LEIST method is widely used to transform a stable atomically precise nanocluster into another atomically precise cluster with a different size and structure through ligand exchange. Konishi et al. [25] reported the reaction of cyclic [Au_9_(PPh_3_)_8_](NO_3_)_3_ with excess dppp (dppp = Ph_2_P(CH_2_)_3_PPh_2_) at room temperature, resulting in the formation of an edge-sharing triangular bipyramid [Au_8_(dppp)_4_](NO_3_)_2_ with a different size and structure. Metallic nanoclusters with multidimensional self-assembled structures have been formed by connecting organic ligands and halide atoms as linkers. They have recently been used to tailor the physicochemical properties [26]. Zhu et al. [27] synthesized three-dimensional metal nanoclusters using [Au_1_Ag_22_(SAdm)_12_](SbF_6_)_2_Cl and [Au_1_Ag_22_(SAdm)_12_](SbF_6_)_3_ as building blocks.

In addition, the irreversible structural isomerization of atomically precise nanoclusters has been reported in a few cases, while the reversible conformational isomerization of metal nanoclusters has been rarely explored [27]. Teo et al. [28,29,30] reported the thermal transformation of metastable Au_38_(Sc_2_H_4_PH)_24_ to the thermodynamically stable icosahedral Au_38_(Sc_2_H_4_PH)_24_ under elevated temperature conditions. However, it is unfortunate that the reverse transformation does not occur. Chen et al. [31] achieved reversible conversion between Au_28_(Sc_6_H_11_)_2_ and Au_28_(SpHC_4_H_9_)_20_ nanoclusters. However, this case involves different surface-protecting ligands (-SC_6_H_11_ and -SC_4_H_9_), which does not meet the definition of isomeric transformation.

In this review, we report on the synthesis and characterization of metal nanoclusters with precise structures, with a particular focus on the relationship between structure and properties and the driving forces behind their synthesis. Due to the differences in atomic composition and synthetic approaches, we discuss the physical and optical properties of metal nanomaterials separately. Finally, we provide an outlook on the future developments in this field.

### 1.2. Definition of Dynamic Metal Nanoclusters

Dynamic metal nanoclusters refer to a class of nanoscale metallic assemblies that exhibit dynamic behavior or undergo structural transformations under certain conditions. Unlike conventional static nanoclusters, dynamic metal nanoclusters can exhibit changes in their size, shape, composition, or ligand environment, leading to altered properties and functionalities. These dynamic characteristics arise from the inherent flexibility and adaptability of the metal nanocluster structures, allowing them to respond to external stimuli or undergo reversible transformations. The study of dynamic metal nanoclusters has gained significant attention due to their unique properties and potential applications in various fields, including catalysis, optics, electronics, and energy storage.

## 2. Advances in the Synthesis and Properties of Dynamic Metal Nanoclusters

The synthesis and properties of dynamic metal nanoclusters have seen significant progress in recent years. Researchers have developed various methods to synthesize these nanoclusters with precise control over their size, composition, and surface properties. These synthesis techniques include ligand exchange, surface modification, templated growth, and size conversion strategies [32,33,34,35,36,37,38]. Researchers have discovered that the band structure of ultrafine metal nanocrystals may differ from that of bulk metals and larger metal nanoparticles. This leads to changes in the surface thermodynamics of metal particles, resulting in different catalytic activity and physical properties. The synthesis of gold nanoparticles protected by a monolayer of thiol ligands using the two-phase and single-phase methods has provided a convenient and effective approach for size-controlled preparation of metal nanoclusters, greatly facilitating their subsequent development. Other research groups, both domestically and internationally, have also reported on the synthesis and properties of various transition metal nanoclusters, including platinum, palladium, silver, copper, and alloys [39,40,41,42,43].

The study of dynamic metal nanoclusters has revealed intriguing phenomena, including size-dependent catalytic activity, enhanced plasmonic effects, and tunable emission properties. These findings have opened up new avenues for applications in catalysis, sensing, imaging, and drug delivery. Furthermore, the understanding of the structure–property relationships in dynamic metal nanoclusters has provided insights into the fundamental principles governing their behavior.

Overall, the synthesis and properties of dynamic metal nanoclusters continue to be an active area of research, offering great potential for the development of novel materials with tailored functionalities. Further investigations into their synthesis methodologies, characterization techniques, and applications are expected to advance our understanding and utilization of these fascinating nanoscale systems.

### 2.1. Synthesis and Characterization of Au_13_Ag_12_(PPh_3_)_10_Cl_8_ Nanoclusters (Single-Atom Exchange)

The synthesis and characterization of Au_13_Ag_12_(PPh_3_)_10_Cl_8_ nanoclusters through single-atom exchange have been investigated. These nanoclusters are composed of a combination of gold (Au) and silver (Ag) atoms, with a ligand shell consisting of PPh_3_ (triphenylphosphine) and chloride (Cl) ions. With regard to the synthesis of Au_13_Ag_12_(PPh_3_)_10_Cl_8_ nanoclusters, Qin et al. [44] reported the formation of Au_13_Ag_12_·Au_12_Ag_13_ “pigeon pair” clusters through single-atom exchange between Au_13_Ag_12_ and AgCl. TGA (thermogravimetric analysis), ESI-MS (electrospray ionization mass spectrometry), and SCXRD (single-crystal X-ray diffraction) confirmed the successful replacement of individual Au atoms in Au_13_Ag_12_ with single Ag atoms to form Au_12_Ag_13_ clusters. However, the location of the thirteenth Ag atom is still unknown. SCXRD revealed the crystallization of the clusters in the space group of p_bca_. The results reported in the literature are shown in Figure 1. In the transformation process, half of the Au and Ag atoms in the M_core_ undergo conversion. During the transformation from Au_13_Ag_12_ to Au_12_Ag_13_, weak Ag-Cl bonds exhibit stronger kinetic properties, potentially serving as possible sites for Ag-Au exchange, likely due to the electrochemical potential difference between Ag and Au.

Through DFT (density functional theory) simulations, the mechanism and pathways of single-atom exchange were investigated. The structural optimization of all structures was performed using the Perdew, Burke, and Ernzerhof (PBE) method [45] combined with the double numeric polarization (DNP) basis set and the Dmol3 program [46]. Self-consistent calculations were carried out to obtain the total energy, with a convergence criterion of 10^−^^5^ Hartree. The UV-vis spectrum of [Au_13_Ag_12_(PH_3_)_10_Cl_8_]^+^ was calculated using the PBE functional [47], with the 6–31G* full-electron basis set for H, P, and Cl, and the LANL2DZ effective core potential basis set for Au and Ag. The results reported in the literature are depicted in the figure below. DFT simulations revealed two possible reaction pathways for single-atom exchange in [Au_13_Ag_12_(PPh_3_)_10_Cl_8_]^+^ clusters, Figure 2. In the [Au_13_Ag_12_(PH_3_)_10_Cl_8_]^+^ cluster, the introduction of an Ag^+^ cation at the waist region leads to the formation of a quasi-linear structure consisting of three Ag atoms and one Au atom. This arrangement undergoes a linear shuttle reaction. Following this reaction, two pathways emerge. Pathway 1 involves a reverse linear shuttle reaction similar to the previous step, resulting in the detachment of the introduced Ag from the surface. In pathway 2, there is no reverse shuttle process between the three Ag atoms and one Au atom. Instead, a linear shuttle occurs, where Ag(3) detaches from the surface. Upon introducing another Ag^+^ cation and undergoing a linear shuttle process, the Au atom detaches from the surface, resulting in the formation of [Au_12_Ag_13_(PPh_3_)_10_Cl_8_]^+^. In the [Au_13_Ag_12_(PH_3_)_10_Cl_8_]^+^ cluster, there are two possible reaction pathways for single atom exchange reactions. This provides a better understanding of the formation of the “{[Au_13_Ag_12_(PPh_3_)_10_Cl_8_]^+^·[Au_12_Ag_13_(PPh_3_)_10_Cl_8_]^+^}” pigeon pair clusters.

### 2.2. Synthesis and Characterization of [Au_25−y_Ag_y_(PPh_3_)_10_Cl_8_]^+^ Nanoclusters (Photoinduced)

The ligand exchange-induced size/structure transformation (LEIST) method has been widely used to transform a stable, atomically precise nanocluster (precursor) into another atomically precise nanocluster through ligand exchange. Photoinduced methods, on the other hand, utilize light as an energy source to excite metal ions, leading to the formation of cores and nanoclusters. Qin et al. [48] reported the synthesis of [Au_37−x_Ag_x_(PPh_3_)_13_Cl_10_]^3+^ (M37) nanoclusters through the reduction of Ph_3_PAuCl with Ph_3_PAgCl using NaBH4 in anhydrous ethanol. M37 nanoclusters undergo size/structure transformation under light irradiation. The irreversible size/structure transformation from M37 to M25 clusters was observed in situ using time-dependent UV-vis, ESI-MS, and femtosecond transient absorption spectroscopy. The M25 cluster was expected to exhibit a rod-like framework with a dodecahedral M25 metal core. The results from the literature are depicted in Figure 3. The DFT results indicate that the weak bond between μ_3_Cl and the Ag_cap_ of the M37 cluster dissociates under light irradiation, resulting in the formation of [M_36_(Ph_3_P)_12_Cl_10_]^2+^ species, which then rapidly converts into M25 and other smaller metal clusters, such as [Au_2_Ag(Ph_3_P)_2_Cl]^+^ and [AuAg_5_(Ph_3_P)_2_Cl_4_]^+^. These free [AgPh_3_P]^+^ ions or small clusters further undergo reduction to Ag(0) and Au(0) under light irradiation and adhere to the reactor wall in the form of Ag or Au mirrors.

### 2.3. Synthesis and Characterization of [Au_13_Ag_12_(PPh_3_)_10_Cl_8_]SbBF_6_ Nanoclusters (Reduction Method)

The Au_13_Ag_12_ nanoclusters were synthesized by reducing a mixture of Ph_3_PAuCl and AgSbF_6_ using NaBH_4_ in an ice bath [49]. The synthesized nanoclusters were analyzed using electrospray ionization mass spectrometry (ESI-MS) in positive mode, and their molecular formula was determined to be [Au_13_Ag_12_(PPh_3_)_10_Cl_8_]^+^, indicating the high purity of the product. Thin-layer chromatography (TLC) identified the presence of two isomers in the Au_13_Ag_12_ product. Fresh solutions containing both isomers of Au_13_Ag_12_ nanoclusters were found to exclusively contain S-Au_13_Ag_12_ when kept at 25 °C for 4 weeks, and E-Au_13_Ag_12_ when kept at −10 °C for 6 weeks. The results from the literature are shown in the Figure 4. The process was monitored using UV-visible absorption spectroscopy, and both E-Au_13_Ag_12_ and S-Au_13_Ag_12_ isomers were obtained with 100% selectivity at −10 °C and 25 °C. ESI-MS confirmed the stability of both isomers, and the isomer conversion process exhibited typical phase transitions. 

### 2.4. Synthesis and Characterization of Multidimensional Silver Cluster-Based Polymers (Ag-CBPs) via Self-Assembly

In this study [50], we describe the synthesis and characterization of one-dimensional {[Ag_22_(L1)_8_(CF_3_CO_2_)_14_](CH_3_OH)_2_}_n_ chains and two-dimensional {[Ag_12_(L2)_2_(CO_2_CF_3_)_14_(H_2_O)_4_(AgCO_2_CF_3_)_4_](HNEt_3_)_2_}_n_ sheets, which were constructed through the bottom-up assembly of Ag_22_-CBPs and Ag_16_-CBPs, Figure 5. Alkanoic acids and thiolate salts were selected as ligands due to their strong interactions with Ag atoms and flexible coordination capabilities, resulting in highly stable Ag-CBPs. The composition and atomic structures of the Ag-CBPs were determined using single-crystal X-ray diffraction. The Ag_22_-CBPs consist of asymmetric units of Ag_22_-CBP, which rotate 180° around the c2 axis to form Ag_22_-CBP monomers. These monomers are connected head-to-tail via Ag-Ag bonds, Ag-O-C (CF_3_) -O-Ag and Ag-trifluoroacetate-Ag sequences, as well as Ag-alkanoic acid bonds, forming one-dimensional silver chains along the c-axis. The Ag_16_-CBPs are formed by the interconnection of two Ag_6_ units, resulting in the formation of zigzag Ag_12_ clusters along the z-axis. The Ag_12_ clusters are connected head-to-tail along the b-axis via (Ag_12_)-trifluoroacetate-(Ag_12_) sequences, and each Ag_12_ cluster interacts with four neighboring Ag_12_ clusters through four Ag_2_ units via (Ag_12_)-trifluoroacetate-(Ag_2_) sequences, forming a two-dimensional sheet structure.

## 3. Properties and Applications of Dynamic Metal Nanoclusters

Dynamic metal nanoclusters possess unique properties that make them highly attractive for various applications. These nanoclusters are characterized by their small size, typically consisting of a few to a few dozen metal atoms, and their ability to undergo dynamic structural changes in response to external stimuli or environmental conditions. Here, we will discuss the properties and applications of dynamic metal nanoclusters.

### 3.1. Catalytic Applications

The electronic structure of Au_12_Ag_13_ clusters is significantly perturbed by the exchange of Ag atoms, resulting in differences in catalytic performance. Both Au_13_Ag_12_ and Au_13_Ag_12_·Au_12_Ag_13_ clusters were loaded on TiO_2_ for the photocatalytic conversion of ethanol [44]. Under UV irradiation (λ = 365 nm) at 30 °C, the conversion rate of ethanol by Au_13_Ag_12_·Au_12_Ag_13_ clusters was 1.5 times higher than that of Au_13_Ag_12_ clusters. The selectivity of Au_13_Ag_12_·Au_12_Ag_13_ clusters towards ethanol was slightly higher compared to Au_13_Ag_12_ clusters. The single-atom exchange (Ag) in M25 clusters with different electronic properties indeed has a significant impact on catalytic activity. In their study, Kauffman et al. [51] investigated the catalytic performance of Au_25_ clusters with precise structures and discovered a reversible process between Au_25_ clusters and CO_2_. The reversible Au_25_-CO_2_ interaction led to changes in spectroscopic and electrochemical properties, which were attributed to the induced redistribution of charges within the clusters upon CO_2_ binding, Figure 6. This spontaneous self-coupling recognition resulted in the application of Au_25_ as an electrochemical catalyst for the reduction of CO_2_ in aqueous media. Dynamic metal nanoclusters have shown great potential in catalytic applications. Their high surface-to-volume ratio, unique electronic structure, and dynamic nature make them efficient catalysts for various chemical reactions, including hydrogenation, oxidation, and carbon dioxide reduction. The ability of nanoclusters to undergo structural changes during catalytic processes enhances their catalytic activity and selectivity.

Li et al. [52] conducted a study on the catalytic performance of water-soluble Au_n_(SR)_m_ nanocluster catalysts [Au_15_(SG)_13_, Au_18_(SG)_14_, Au_25_(SG)_18_, Au_38_(SG)_24_, and Au_25_(Capt)_18_] in the homogeneous chemical selective hydrogenation reaction in water. They observed significant size dependence and spatial effects of ligands in the hydrogenation reactions catalyzed by gold nanoclusters. The catalytic activity of the gold nanoclusters (based on the conversion of 4-nitrobenzaldehyde) increased with increasing core size: Au_15_(SG)_13_ < Au_18_(SG)_14_ < Au_25_(SG)_18_ < Au_38_(SG)_24_. On the other hand, gold nanoclusters with smaller ligand volumes exhibited better catalytic performance [Au_25_(Capt)_18_ > Au_25_(SG)_18_]. The DFT calculations were performed using the Perdew–Burke–Ernzerhof (PBE) functional [45] based on the generalized gradient approximation, the def2-SVP basis set, and empirical dispersion correction. The surface area of the clusters in the solvent was calculated using the default values of r_i_ and r_solv_ in Turbomole 6.5 [53]. The results showed that both the -CHO and -NO_2_ groups of the 4-nitrobenzaldehyde molecule interacted closely with the surface bonds (S-Au-S) of the clusters. The adsorption energies of 4-nitrobenzaldehyde on four different-sized Au_n_(SR)_m_ nanoclusters ranged from −0.86 to −1.05 eV. The increased catalytic activity of larger gold nanoclusters was consistent with the surface area of the nanoclusters. Insights into the molecular-level understanding of the hydrogenation of nitrobenzaldehyde and the catalytic active site structure on gold nanocluster catalysts were revealed, Figure 7.

### 3.2. Optical Applications

The electronic structure of metal nanoclusters greatly influences their fluorescence properties. The maximum excitation wavelength of Au_13_Ag_12_·Au_12_Ag_13_ clusters is increased compared to Au_13_Ag_12_, and a redshift phenomenon is observed in the fluorescence spectrum. The quantum yield (QY) is significantly enhanced. Metal nanoclusters find wide applications in areas such as lighting, biotechnology, and medicine [44]. Pniakowska et al. [54] conducted experimental research and time-dependent density functional theory (TD-DFT) simulations to investigate the influence of gold atom doping on the single-photon and two-photon absorption and emission properties of connected silver nanoclusters. They observed that central gold doping had a significant impact on the linear and nonlinear optical properties, resulting in enhanced luminescence quantum yield and two-photon intensity, Figure 8. They also predicted that AuAg clusters with precise structures could serve as luminescent probe materials.

### 3.3. Electronics Applications

Due to the discovery that the two enantiomers of metal nanoclusters (S-Au_13_Ag_12_ and E-Au_13_Ag_12_) can undergo reversible conversion by controlling the temperature [49], it has been found that the metal configuration with higher symmetry (D5h) is favored for the formation of the E-Au_13_Ag_12_ isomer at low temperatures (−10 °C). As the temperature increases to 25 °C, the S-Au_13_Ag_12_ isomer (lower symmetry) is exclusively formed. Alloying and ligand engineering (such as Ag–halogen bonding) provide a rational strategy to make the framework of metal nanoclusters more flexible, enabling conformational isomerism with the potential for designing smart molecular engines driven by Gibbs free energy. These temperature-driven, interconvertible nanocluster isomers open up avenues for the design of thermal sensors and intelligent catalysts utilizing ultra-small nanoclusters. 

Unlike common silver complexes, silver clusters, or even silver wires, here we obtained the silver frameworks in Ag_22_-CBP, silver chains, and silver disks in Ag_16_-CBP, which possess unique and continuous features, providing favorable pathways for electron transport and enabling us to investigate their conductivity in the solid state. The conductivity of Ag_22_-CBP and Ag_16_-CBP changes upon contact with methanol and ethanol, while it remains insulated when in contact with acetone, toluene, and ultrapure water. Both Ag_22_-CBP and Ag_16_-CBP serve as excellent sensors for protonic organic solvents such as methanol and ethanol. As shown in Figure 9. Ag-CBPs exhibit high electrical response towards methanol and ethanol, suggesting that cluster-based materials can be used for designing sensitive sensors for detecting organic solvents [50]. Dynamic metal nanoclusters have demonstrated excellent sensing capabilities. They can be engineered to respond to specific analytes or environmental changes, leading to highly sensitive and selective detection of various substances, including gases, ions, biomolecules, and pollutants. This makes them promising candidates for applications in environmental monitoring, medical diagnostics, and food safety.

Yonesato et al. [55] developed precise structures of Ag nanoclusters with polyoxometalate (POMs) as inorganic ligands. These metal nanoclusters exhibited stimulating and reversible electronic state responses, which was highly attractive. As shown in Figure 10, through protonation/deprotonation of the surrounding POMs ligands, the electronic state of a precisely defined {Ag_27_} nanocluster was able to be reversibly controlled at the atomic level. This manipulation of the cluster’s electronic state was accompanied by significant changes in the UV-visible absorption spectrum. The study confirmed the enormous potential of Ag nanoclusters in the field of electronics, particularly in sensing applications.

## 4. Trends and Challenges in Dynamic Nanometal Clusters

In summary, dynamic metal nanoclusters possess fascinating properties that make them versatile materials for a wide range of applications. Their tunable properties, catalytic activity, optical application, electronics applications, and sensing abilities make them promising candidates for future technological advancements in various fields. Dynamic nanometal clusters have gained significant attention in recent years due to their unique properties and potential applications. These clusters, consisting of a small number of metal atoms, exhibit fascinating size-dependent behavior and can undergo structural transformations in response to external stimuli. Here, we discuss the trends and challenges associated with the development of dynamic nanometal clusters.

### 4.1. Size and Composition Control

One of the major trends in the field is the precise control over the size and composition of nanometal clusters. In terms of size control, the complex structure and atomic-scale variations of dynamic nanoclusters make it difficult to precisely manipulate their sizes. Conditions during the synthesis process, reaction kinetics, and aggregation states can all influence the cluster sizes. Additionally, controlling the composition of dynamic nanoclusters also presents challenges. In the synthesis process, precise control over the selection, ratio, and doping of different metal atoms is required to achieve the desired composition.

These challenges stem from several factors. Firstly, the choice and optimization of synthesis methods are crucial for achieving precise size and composition control. Methods that can effectively control reaction kinetics and product composition need to be developed. Secondly, a deep understanding of the relationship between the structure and properties of dynamic nanoclusters is necessary. This involves the development of structural analysis and property characterization techniques. Finally, the thermodynamics and kinetics involved in the synthesis process need to be carefully studied to achieve precise size and composition control.

### 4.2. Structural Characterization

Accurate structural characterization of dynamic nanometal clusters is crucial for understanding their behavior and properties. Advanced characterization techniques, such as X-ray diffraction (XRD), Fourier transform infrared spectroscopy, electrospray ionization mass spectrometry (ESI-MS), thermogravimetric analysis (TGA), UV−vis spectroscopy, and spectroscopic methods are being employed to determine the atomic arrangement and study the structural transformations of these clusters.

The structural characterization of dynamic metal nanoclusters faces several challenges in practice:

(a) There are challenges in measuring the dynamic phenomena in dynamic metal nanoclusters. The dynamic processes of metal nanoclusters typically occur on the femtosecond or picosecond timescale. Accurate measurement and observation of these processes require experimental equipment and techniques with high temporal resolution, such as femtosecond laser systems and fast detectors. However, due to the short timescale of the dynamic processes in metal nanoclusters, conventional experimental techniques often cannot directly observe these processes. Therefore, the key to explaining and understanding the observed phenomena is to combine experimental results with theoretical models. Computational simulations can be used to predict and validate the experimental results, helping to explain the dynamic behavior of the clusters. This integrated approach of combining experimental and theoretical methods can provide a deeper understanding of the behavior of dynamic metal nanoclusters. The combination of experimental and theoretical methods is an effective approach for studying the behavior of dynamic metal nanoclusters. By integrating experimental results with theoretical models, a more comprehensive understanding and explanation of the observed phenomena can be achieved, leading to deeper insights into the behavior of dynamic metal nanoclusters [56,57,58,59].

(b) Dynamic metal nanoclusters exhibit complex structural features, including variations in atomic composition, crystal structure, and surface ligands. Due to their small size and structural diversity, conventional structural characterization techniques may not directly observe their precise structures. Therefore, a combination of multiple characterization methods is required for comprehensive analysis.

(c) Dynamic metal nanoclusters undergo reversible isomerization, where their structures and properties can change with varying environmental conditions. This dynamic nature makes it challenging to accurately understand their structures, necessitating the use of time-resolved experiments and theoretical simulations to reveal their dynamic behavior.

(d) Size effects are significant in dynamic metal nanoclusters [60,61,62,63], where their structures and properties can be influenced by their sizes. However, precise control over the size of nanoclusters remains a challenge, especially for synthesizing and characterizing large-sized clusters.

(e) Surface modification is commonly employed in dynamic metal nanoclusters through ligand attachments, which can impact their structures and properties. However, the ligand environment on the cluster’s surface is often difficult to directly observe and determine, requiring the utilization of surface analysis techniques and theoretical simulations to uncover their structures and properties.

(f) In practical applications, dynamic metal nanoclusters may exhibit clustering phenomena, where multiple clusters aggregate together [64,65,66]. Cluster aggregation can affect the structures and properties of the individual clusters, posing challenges in characterizing and analyzing the structure of clustered systems.

To address these challenges, a combination of advanced characterization techniques is necessary to gain a comprehensive understanding of the structure–property relationship of dynamic metal nanoclusters. Additionally, the development of innovative characterization methods is crucial for overcoming these challenges.

### 4.3. Functional Applications

Dynamic nanometal clusters hold great potential for various applications. They can be utilized in catalysis, sensing, energy storage, and biomedical fields. The ability of these clusters to undergo reversible structural changes opens up new opportunities for developing responsive materials and devices with enhanced performance.

Despite the promising prospects, several challenges exist in the field of dynamic nanometal clusters. In the field of catalysis, the challenge lies in the design and synthesis of clusters with ideal catalytic performance, including controlling their sizes, compositions, and structures, as well as improving their stability and regenerability. In optical applications, precise control over the morphologies, sizes, and compositions of clusters is needed to achieve specific optical responses and improve photostability and quantum yield. Finally, in the field of sensing, the challenge is to design sensors with higher sensitivity and selectivity, and to achieve specific interactions with target molecules.

### 4.4. Stability and Scalability

Ensuring the stability of dynamic clusters under different conditions and scaling up their synthesis remain challenging tasks. Developing robust synthetic methods and stabilizing agents is necessary to maintain the desired properties of these clusters in practical applications.

There are many challenges in stability and scalability. Dynamic metal nanoclusters have the characteristics of reversible isomerization and structural rearrangement, which make the stability of clusters difficult to predict and control. Under different environmental conditions, the structures and properties of clusters may change, resulting in their stability being affected. The surfaces of clusters are usually covered by ligands or modified molecules, and these surface modifications play a key role in the stability of clusters. However, the selection and control of surface modification is a challenge because different ligands or modified molecules have different effects on the stability and activity of clusters. Large-scale preparation of dynamic metal nanoclusters with consistency and controllability is a challenge. The synthesis method needs to be reproducible to ensure that different batches of clusters have similar properties. At the same time, the synthesis method also needs to be controllable in order to control the sizes, morphologies, and compositions of the clusters. Cost-effectiveness and resource availability need to be considered to achieve scalable applications of dynamic metal nanoclusters. Some metal atoms or ligands may be expensive or scarce, limiting the large-scale preparation and application of clusters. 

### 4.5. Understanding the Structure–Property Relationship

Establishing a comprehensive understanding of the relationship between the atomic structures and the properties of dynamic nanometal clusters is crucial for their rational design [67]. This requires further theoretical and experimental investigations to elucidate the underlying principles governing their behavior. The understanding of the structure–property relationship in dynamic metal nanoclusters can provide valuable insights into their functionality and potential applications. However, there are challenges in unraveling this relationship. Firstly, for complex structures, it is sometimes difficult to directly observe and determine their precise structure using traditional characterization techniques. This challenge is particularly pronounced for larger clusters or those present in solutions. Secondly, the dynamic nature of metal nanoclusters, which undergo reversible isomerization, leads to variations in their structures and properties under different environmental conditions, such as temperature, solvent, and atmosphere. This dynamic behavior adds complexity to the understanding of the structure–property relationship and requires a combination of experimental and theoretical studies to reveal their dynamic behavior. Lastly, the selection of different metal elements, ligands, and synthesis methods significantly influences the structures and properties of dynamic metal nanoclusters. The diversity and variability make it challenging to establish a universally applicable structure–property relationship model.

### 4.6. Integration and Compatibility

Integrating dynamic nanometal clusters into practical devices and systems while maintaining their dynamic behavior poses challenges. Compatibility with different matrices and substrates, as well as long-term stability, need to be addressed for successful integration into functional devices. In general, dynamic metal nanoclusters have vast application prospects, but they also require overcoming various challenges encountered during their preparation and application processes. These challenges include improving preparation efficiency and simplifying synthesis steps, optimizing assembly materials, studying the relationship between the structure and properties of dynamic metal nanoclusters, and enhancing their evaluation in environmental and biological contexts. Only by addressing these challenges can the application and commercialization of dynamic metal nanoclusters be realized in different fields.

## 5. Conclusions

Through the review of dynamic metal nanoclusters, it is evident that they are a fascinating class of nanomaterials with unique properties and potential applications. Comprehensive experimental and theoretical studies have deepened our understanding of the synthesis, structures, properties, and behaviors of dynamic metal nanoclusters. Optimized synthesis methods and control strategies have enabled precise control over the morphologies, compositions, and structures of metal nanoclusters, laying a solid foundation for subsequent application research. Characterization techniques for crystal structures and theoretical simulations have facilitated the design and optimization of functional cluster materials. Dynamic metal nanoclusters exhibit promising potential in catalysis, optics, electronics, and energy storage. By manipulating the size, composition, and surface structures of clusters, efficient catalysts and optoelectronic materials can be designed and synthesized for various chemical reactions and energy conversion processes. Furthermore, dynamic metal nanoclusters play a significant role in materials science and nanotechnology. However, future research on dynamic metal nanoclusters will focus on precise synthesis and control, the development of characterization techniques, theoretical simulations and calculations, advancement of practical applications, and considerations of environmental and safety aspects. These efforts aim to further expand their applications in catalysis, optics, electronics, and sensing, while addressing related challenges.

## Figures and Tables

**Figure 1 molecules-28-05306-f001:**
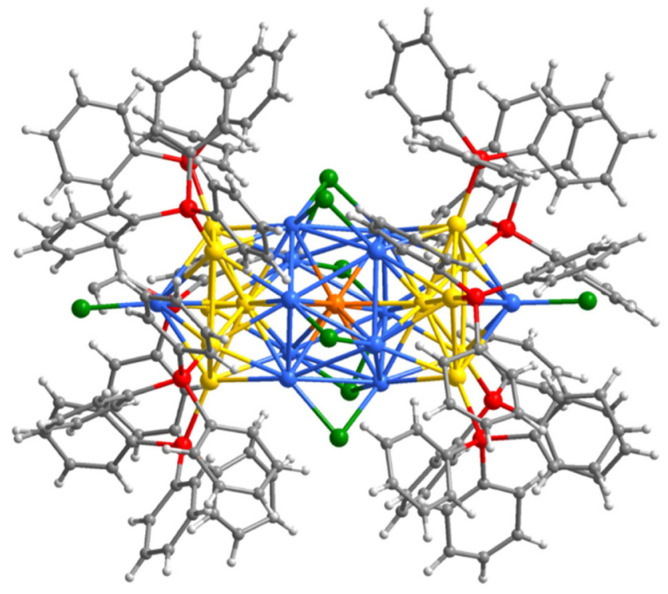
Crystal structure of Au_12_._5_Ag_12_._5_ nanoclusters. Color code: Au, yellow; Ag, light blue; Au/Ag, orange; Cl, green; P, red; C, gray; H white. Reproduced with permission from Ref. [40]. Spring 2022.

**Figure 2 molecules-28-05306-f002:**
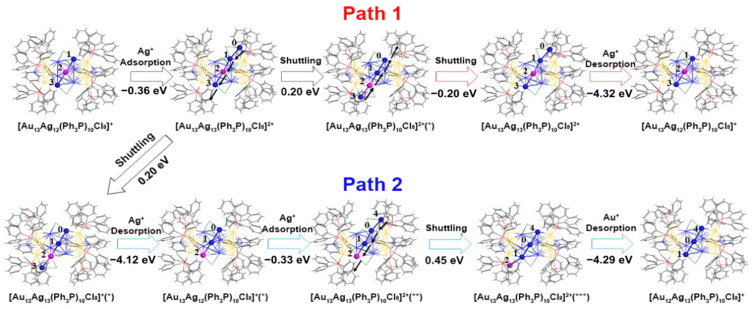
DFT-calculated mechanisms for the Ag or Au exchange between the Au_13_Ag_12_ and Au_12_Ag_13_ clusters. Reproduced with permission from Ref. [40]. Spring 2022. * *p* ≤ 0.05; ** *p* ≤ 0.01; *** *p* ≤ 0.001.

**Figure 3 molecules-28-05306-f003:**
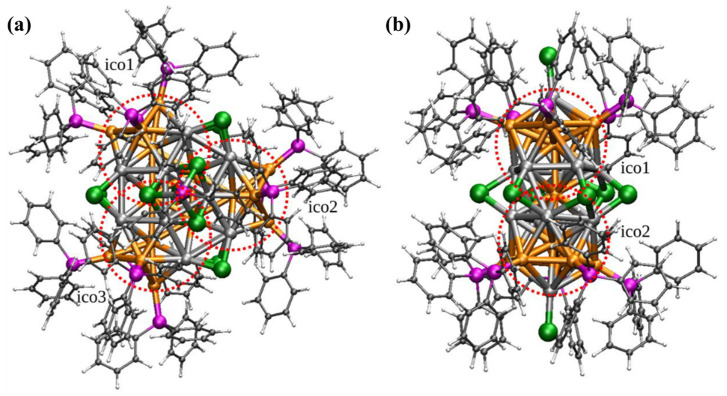
DFT-optimized structures of (**a**) M37 and (**b**) M25 clusters. The local icosahedral components are marked by “ico#”. Color code: M, orange; Cl, green; P, pink; C, black; H white. Reproduced with permission from Ref. [41]. American Chemical Society 2021.

**Figure 4 molecules-28-05306-f004:**
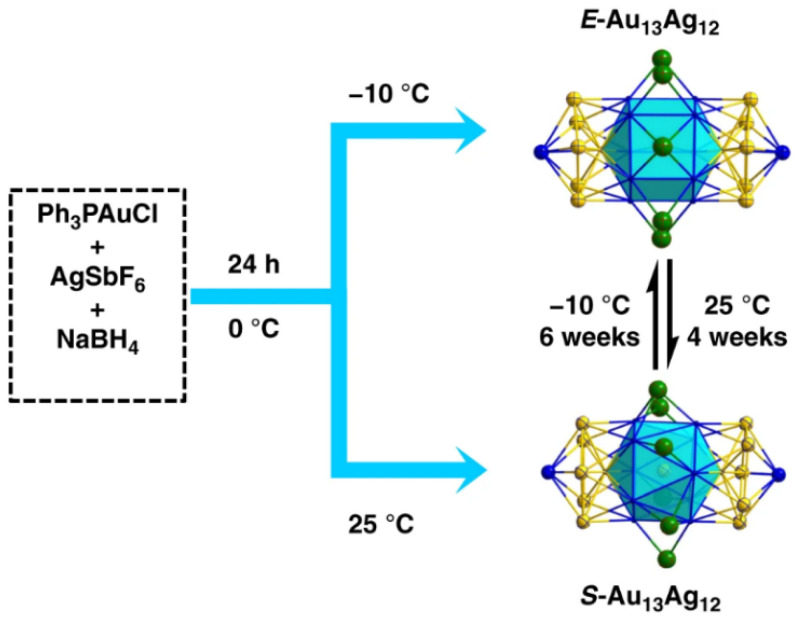
Two isomers of the [Au_13_Ag_12_(PPh_3_)_10_Cl_8_]^+^(SbF_6_)^−^ nanocluster with thermally responsive transformation. E- and S- stand for eclipsed and staggered configurations. Color code: Au, yellow; Ag, blue; Cl, green. Reproduced with permission from Ref. [42]. Nature Group 2020.

**Figure 5 molecules-28-05306-f005:**
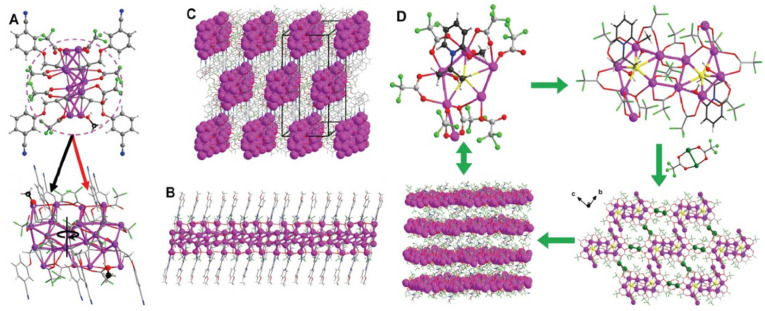
(**A**) The asymmetric structure and monomer of Ag_22_-CBP. (**B**) The 1D silver chain along the c axis. (**C**) The 3D structure with holes. (**D**) Structural anatomy of Ag_16_-CBP: Ag6 unit in asymmetry, Ag_12_ unit in monomer and 2D and 3D structures. Color code: Ag, purple and green; S, yellow; F, cyan; O, red; N, blue; C, gray and black; H, white. Reproduced with permission from Ref. [50]. Science Group 2022.

**Figure 6 molecules-28-05306-f006:**
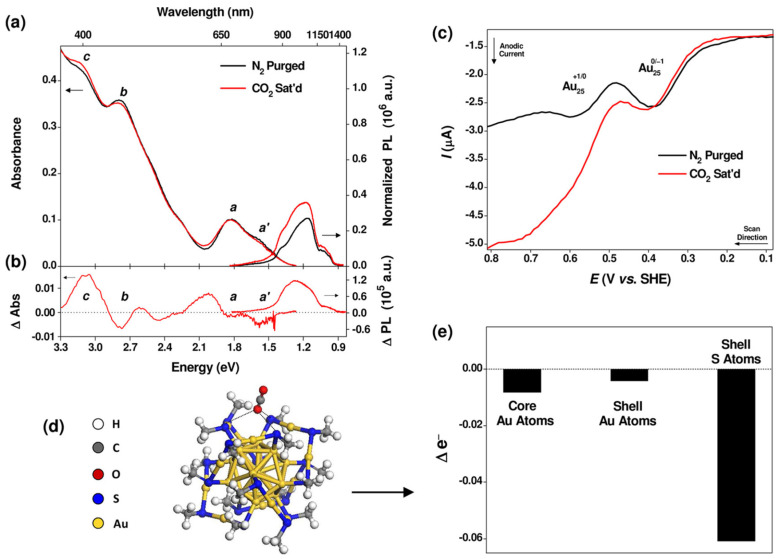
(**a**) Optical absorbance and PL spectra, and (**b**) difference spectra of Au_25_ in N_2_−purged and CO_2_-saturated DMF. (**c**) Square wave voltammetry of Au_25_ in N_2_-purged and CO_2_−saturated DMF + 0.1 M TBAP. (**d**) DFT model of stable CO_2_ adsorption where an O atom of CO_2_ interacts with three S atoms in the Au_25_ shell. (**e**) Bader charge analysis. Reproduced with permission from Ref. [51]. American Chemical Society 2012.

**Figure 7 molecules-28-05306-f007:**
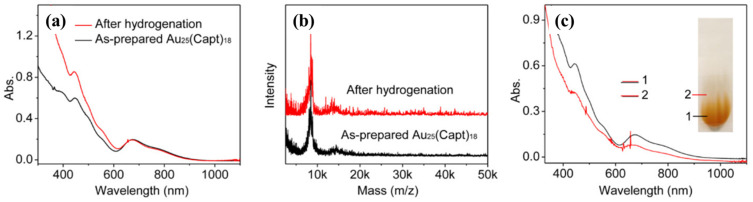
(**a**) UV−vis and (**b**) MALDI mass spectrometry spectra of the Au_25_(Capt)_18_ nanocluster catalyst before and after the hydrogenation reaction. (**c**) Gel images and corresponding UV−vis spectra (at different positions) of the gold nanocluster catalysts after the hydrogenation. Reproduced with permission from Ref. [52]. ACS Catalysis 2014.

**Figure 8 molecules-28-05306-f008:**
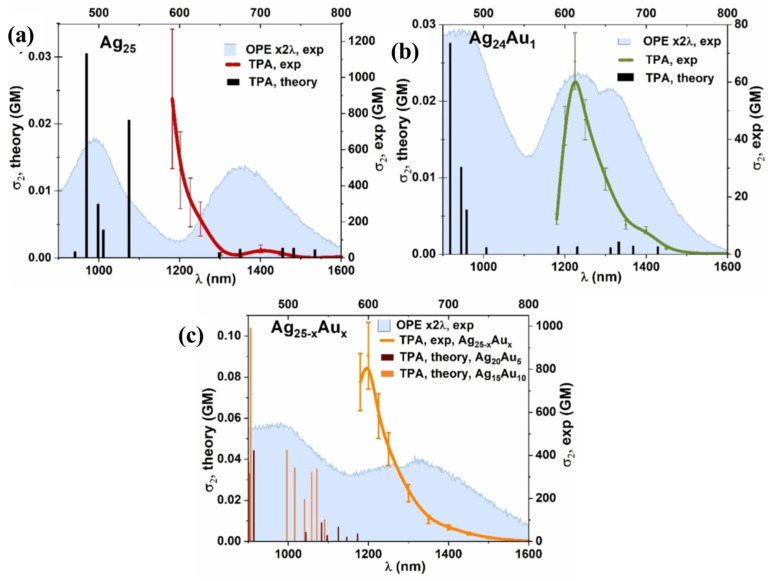
Off-resonance and simulated TPA spectra of (**a**) Ag_25_, (**b**) Ag_24_Au_1_, and (**c**) Ag_25-x_Au_x_ nanoclusters presented with respective normalized one-photon excitation (OPE) at twice the wavelength (blue area). Reproduced with permission from Ref. [53]. John Wiley & Sons 2022.

**Figure 9 molecules-28-05306-f009:**
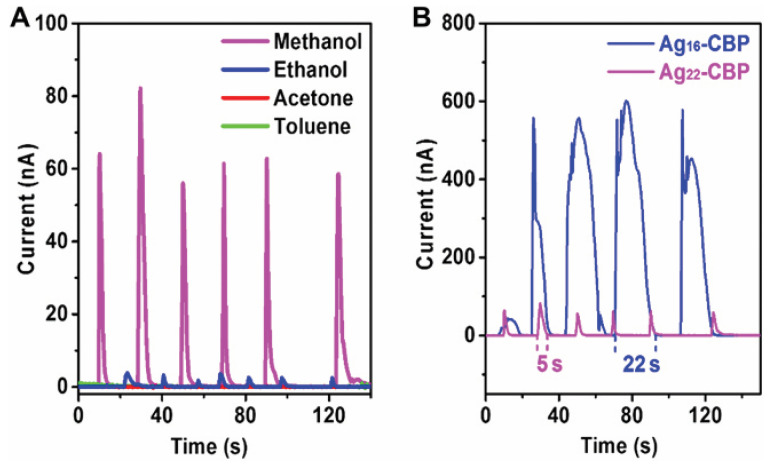
Dynamic response and recovery characterization of Ag-CBP sensors. (**A**) Dynamic response and recovery curves of Ag_22_-CBP thin-film sensors under different organic solvents (protic solvents: methanol and ethanol; aprotic solvents: acetone and toluene). (**B**) Dynamic response and recovery curves of Ag_22_-CBP and Ag_16_-CBP films in the presence of methanol. Reproduced with permission from Ref. [50]. Science Group 2022.

**Figure 10 molecules-28-05306-f010:**
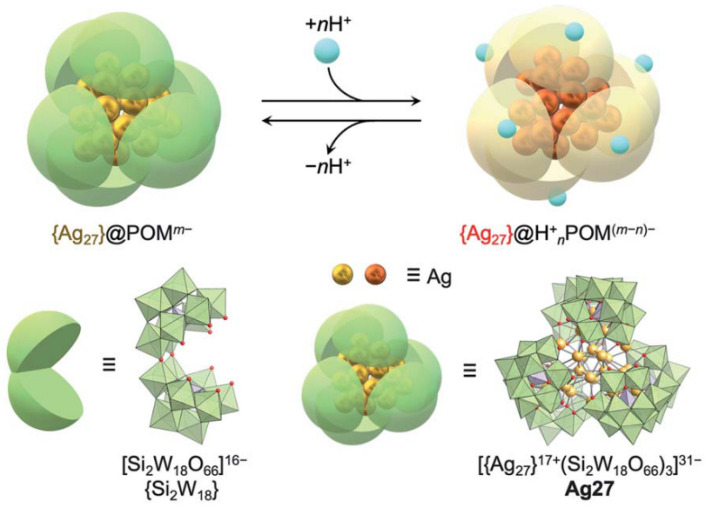
Control of the electronic states of {Ag_27_} nanoclusters via protonation/deprotonation of [Si_2_W_18_O_66_]^16−^. Reproduced with permission from Ref. [54]. Royal Society of Chemistry 2022.

## Data Availability

Not applicable.

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
