# Peer review of "Dynamic Metal Nanoclusters: A Review on Accurate Crystal Structures"

_molecules, 2023, doi:10.3390/molecules28145306_

Round 1

Reviewer 1 Report

This is an interesting review of a specific set of dynamic metal nanoclusters. The authors need to make it far more clear that the review is of very narrow scope focusing on silver-gold nanoclusters in a specific size range. This must be in the title. Th title should state. ….A Review Based on ….. as the focus is not the crystal structures but is broader than just structures.

Given that it is a very narrow class of compounds, the review is good and useful.

There are additional issues to address.

Section 1.1 Dynamic in the first line should not be capitalized.

Section 2. first paragraph. There are repeated sentences here and from the introduction. Eliminate repeats such as this.

Whenever, DFT is mentioned, they need to give the actual DFT method in terms of the functional and basis set as the term DFT tells nothing about the quality of the method.

Make sure to define DFT as density functional theory.

Also, DFT can only predict something. It does not indicate, prove, etc.

 Make sure to give the phase for all DFT calculations. I presume they are all for the gas phase.

As this is a review, titles should be given for all references.

Give references to the DFT methods as well, including the type of TD-DFT done.

In section 4.2, the authors should also discuss issues with measuring dynamic phenomena as the paper is about dynamic nanoclusters. Kinetics of these processes is clearly important and difficult to do but it is not discussed.

These changes must be made for such a review paper as it is not original science.

see above

Author Response

This is an interesting review of a specific set of dynamic metal nanoclusters. The authors need to make it far more clear that the review is of very narrow scope focusing on silver-gold nanoclusters in a specific size range. This must be in the title. Th title should state. ….A Review Based on ….. as the focus is not the crystal structures but is broader than just structures. Given that it is a very narrow class of compounds, the review is good and useful. There are additional issues to address.

Section 1.1 Dynamic in the first line should not be capitalized.

A: Thanks for Reviewer’s comments, we have revised Section 1.1 of the manuscript as:“In recent years, many fundamental properties of dynamic metal nanoclusters have been gradually revealed”.

Section 2. first paragraph. There are repeated sentences here and from the introduction. Eliminate repeats such as this.

A: Thanks for the Reviewer’s comments, we have revised Section 2 of the manuscript as: “The synthesis and properties of dynamic metal nanoclusters have seen significant progress in recent years. Researchers have developed various methods to synthesize these nanoclusters with precise control over their size, composition, and surface properties. These synthesis techniques include ligand exchange, surface modification, templated growth, and size conversion strategies. Researchers have discovered that the band structure of ultrafine metal nanocrystals may differ from that of bulk metals and larger metal nanoparticles. This leads to changes in the surface thermodynamics of metal particles, resulting in different catalytic activity and physical properties. The synthesis of gold nanoparticles protected by a monolayer of thiol ligands using the two-phase and single-phase methods has provided a convenient and effective approach for the size-controlled preparation of metal nanoclusters, greatly facilitating their subsequent development. Other research groups both domestically and internationally have also reported on the synthesis and properties of various transition metal nanoclusters, including platinum, palladium, silver, copper, and alloys”.

Whenever DFT is mentioned, they need to give the actual DFT method in terms of the functional and basis set as the term DFT tells nothing about the quality of the method.

Make sure to define DFT as density functional theory. Also, DFT can only predict something. It does not indicate, prove, etc.

A: Thanks for the reviewer’s comments, we have revised Section 2.1 of the manuscript as:“The structural optimization of all structures was performed using the Perdew, Burke, and Ernzerhof (PBE) method combined with the Double Numeric Polarization (DNP) basis set and the Dmol3 program. Self-consistent calculations were carried out to obtain the to-tal energy, with a convergence criterion of 10-5 Hartree. The UV-vis spectrum of [Au13Ag12(PH3)10Cl8]+ was calculated using the PBE functional in the Gaussian 09 software package, with the 6-31G* full-electron basis set for H, P, and Cl, and the LANL2DZ effective core potential basis set for Au and Ag”.

Make sure to give the phase for all DFT calculations. I presume they are all for the gas phase.

As this is a review, titles should be given for all references.

A: Thanks for the reviewer’s comments. Li et al. studied the catalytic performance of water-soluble Aun(SR)m nanocluster catalysts [Au15(SG)13, Au18(SG)14, Au25(SG)18, Au38(SG)24, Au25(Capt)18] in aqueous solvents. The others are simulated by DFT in gas phase.

Give references to the DFT methods as well, including the type of TD-DFT done.

In section 4.2, the authors should also discuss issues with measuring dynamic phenomena as the paper is about dynamic nanoclusters. Kinetics of these processes is clearly important and difficult to do but it is not discussed.

A: Thanks for the reviewer’s comments, we have revised Section 4.2 of the manuscript as:“There are challenges in measuring the dynamic phenomena in dynamic metal nanoclusters. The dynamic processes of metal nanoclusters typically occur on the femtosecond or picosecond timescale. Accurate measurement and observation of these processes require experimental equipment and techniques with high temporal resolution, such as femtosecond laser systems and fast detectors. However, due to the short timescale of the dynamic processes in metal nanoclusters, conventional experimental techniques often cannot directly observe these processes. Therefore, the key to explaining and understanding the observed phenomena is to combine experimental results with theoretical models. Computational simulations can be used to predict and validate experimental results, helping to explain the dynamic behavior of the clusters. This integrated approach of combining experimental and theoretical methods can provide a deeper understanding of the behavior of dynamic metal nanoclusters”.

These changes must be made for such a review paper as it is not original science.

A: Many thanks for your suggestions, we have revised the review work according to your suggestions.

Reviewer 2 Report

I have carefully reviewed the manuscript "Dynamic Metal Nanoclusters: A Review on Accurate Crystal Structures." The review article comprehensively analyzes the current research on dynamic metal nanoclusters, specifically focusing on accurately determining their crystal structures. I have a few suggestions to improve the quality of the manuscript before it can be considered for publication.

The manuscript is not well-written; there are a few instances where clarity could be improved. I recommend proofreading the manuscript thoroughly to address grammatical errors and improve sentence structure where necessary.

Please provide a clearer roadmap in the introduction, summarizing the sections covered in the review. This will help readers navigate the paper more effectively.

The abstract is very short and does not appeal. Please revise it.

The authors have provided an overview of very few recent research advances; I encourage them to include more up-to-date studies and references to ensure the paper reflects the latest developments in the field.

The review mentions various experimental techniques and computational methods used for crystal structure determination; providing more detailed explanations and examples of their applications would be beneficial. This will enhance the reader's understanding of the techniques and their relevance to dynamic metal nanoclusters.

Please add a few examples of MOF and CP materials with the help of these references.

ACS Applied Materials & Interfaces2023, 15(16), 20064-20074, J. Mater. Chem. C, 2023,11, 3692-3709; Molecules 27, no. 21 (2022): 7166;

The authors briefly mention potential applications of dynamic metal nanoclusters, such as catalysis, optics, electronics, and energy storage. I suggest expanding on these applications and providing specific examples or case studies to illustrate the practical utility of these nanoclusters in each area.

The manuscript would benefit from including figures and tables to visually represent key concepts, experimental setups, or crystal structures discussed in the text. Visual aids will enhance the clarity and impact of the review.

I am not happy with the conclusion part of the review. The paper does not mention future directions and challenges in the field and lacks a concise and well-developed conclusion section. Therefore, suggest expanding on the conclusion and prospects and discussing potential research directions to address the existing challenges.

Author Response

I have carefully reviewed the manuscript "Dynamic Metal Nanoclusters: A Review on Accurate Crystal Structures." The review article comprehensively analyzes the current research on dynamic metal nanoclusters, specifically focusing on accurately determining their crystal structures. I have a few suggestions to improve the quality of the manuscript before it can be considered for publication.

The manuscript is not well-written; there are a few instances where clarity could be improved. I recommend proofreading the manuscript thoroughly to address grammatical errors and improve sentence structure where necessary.

A: Thanks for the reviewer’s comments, we carefully proofread the manuscript to solve grammatical errors and improve sentence structure.

Please provide a clearer roadmap in the introduction, summarizing the sections covered in the review. This will help readers navigate the paper more effectively.

The abstract is very short and does not appeal. Please revise it.

A: Thanks for the reviewer’s comments, In the introduction, we introduce the concept of dynamic metal nanoclusters from the research background of dynamic metal nanoclusters. And we revise the abstract.

The authors have provided an overview of very few recent research advances; I encourage them to include more up-to-date studies and references to ensure the paper reflects the latest developments in the field.

A: Thanks for the reviewer’s comments, we introduce the recent research progress of Ag, Au-based metal nanoclusters. More recent studies and references are also included in section 2.

The review mentions various experimental techniques and computational methods used for crystal structure determination; providing more detailed explanations and examples of their applications would be beneficial. This will enhance the reader's understanding of the techniques and their relevance to dynamic metal nanoclusters.

Please add a few examples of MOF and CP materials with the help of these references.

ACS Applied Materials & Interfaces, 2023, 15(16), 20064-20074, J. Mater. Chem. C, 2023,11, 3692-3709; Molecules 27, no. 21 (2022): 7166;

A: Thanks for the reviewer’s comments, according to your suggestion, we carefully read the recommended three articles and found that it is very necessary to quote.

  1. Jiang, X. L. Li, D. Fang, W. Y. Lieu, C. Chen, M. S. Khan, D.-S. Li, B. Tian, Y. Shi and H. Y. Yang, ACS Applied Materials & Interfaces 15 (16), 20064-20074 (2023).
  2. Shi, Y. Zou, M. S. Khan, M. Zhang, J. Yan, X. Zheng, W. Wang and Z. Xie, Journal of Materials Chemistry C 11 (11), 3692-3709 (2023).
  3. F. Zhang, G. M. Ye, D. H. Liao, X. L. Chen, C. Y. Lu, A. Nezamzadeh-Ejhieh, M. S. Khan, J. Q. Liu, Y. Pan and Z. Dai, Molecules 27 (21) (2022).

The authors briefly mention potential applications of dynamic metal nanoclusters, such as catalysis, optics, electronics, and energy storage. I suggest expanding on these applications and providing specific examples or case studies to illustrate the practical utility of these nanoclusters in each area.

A: Thanks for the reviewer’s comments. In Section 3.1, we increase the application of dynamic metal nanoclusters in catalysis. Catalytic performance of Aun(SG)m nanoclusters catalyst.

The manuscript would benefit from including figures and tables to visually represent key concepts, experimental setups, or crystal structures discussed in the text. Visual aids will enhance the clarity and impact of the review.

A: Thanks for the reviewer’s comments, we have revised Section 4.2 of the manuscript.

I am not happy with the conclusion part of the review. The paper does not mention future directions and challenges in the field and lacks a concise and well-developed conclusion section. Therefore, suggest expanding on the conclusion and prospects and discussing potential research directions to address the existing challenges.

A: Thanks for the reviewer’s comments, in section 5, we add the future directions and challenges in the fields of catalysis, optics, and electronics of dynamic metal nanoclusters. Expand conclusions and prospects and discuss potential research directions to address existing challenges.

Round 2

Reviewer 2 Report

The author has thoroughly revised the manuscript. I am pleased to accept it in its current form.